# Position: Carbon Footprint Reporting Should Be Routine in Machine Learning Research

**Guan-Ming Chiu** [1]

## Abstract

In this position paper, we argue that the machine learning community should adopt standardized carbon footprint reporting as part of routine scientific practice. Training large models can emit hundreds of tons of $CO_2$, yet environmental costs remain largely invisible in publications. We contend that without energy and emissions metrics, claims of model efficiency are incomplete: a method cannot be deemed "efficient" without specifying efficient at what. This gap undermines scientific rigor and reproducibility, as identical experiments in different locations yield vastly different carbon footprints. We put forth reporting guidelines comprising five standardized metrics, practical measurement tools, and integration with community benchmarks, with a phased three-stage adoption process. We address alternative views, including concerns about measurement complexity and potential barriers for resource-limited researchers. To promote equity, we advocate for dual reporting of energy and carbon, reference-grid normalization, and acceptance of approximate estimates. This paper calls on venues, reviewers, authors, and institutions to establish carbon awareness as a foundational element of responsible ML research.

## 1. Introduction

Machine learning progress has a cost our community rarely quantifies: the energy consumed and carbon emitted during training and deployment. A typical ML paper specifies GPU types, training duration, and hyperparameters in detail, yet omits energy consumption or carbon emissions entirely. This asymmetry reflects a blind spot in our scientific culture.

The environmental impact is substantial. Training a Trans-

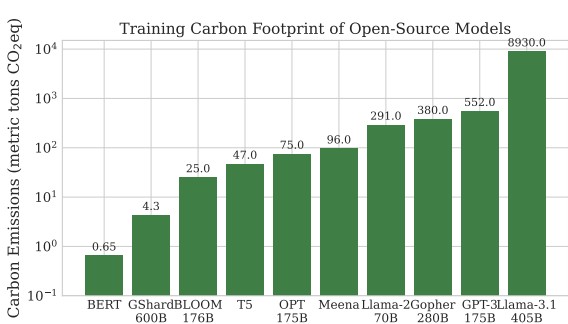

*Figure 1.* Officially published carbon emissions for training major language models (training-phase only).[1] Variation reflects model scale and infrastructure: BLOOM (25 tons) used low-carbon French grid; Llama-3.1 405B is the largest disclosed at 8,930 tons (the full Llama-3.1 family of 8B/70B/405B totals 11,390 tons). Methodologies vary; direct comparisons require caution.

former model with neural architecture search can emit as much carbon as five cars over their lifetimes (Strubell et al., 2019). Computational requirements have increased roughly $10\times$ every two years (Sevilla et al., 2022), and data centers now consume 1–2% of global electricity (Masanet et al., 2020; Xiao et al., 2025).

As shown in Figure 1, emissions vary dramatically: GPT-3 emitted 552 tons (Patterson et al., 2021), while BLOOM achieved comparable scale at just 25 tons using France's nuclear-powered grid (Luccioni et al., 2023). This variation demonstrates how much infrastructure choices matter, yet most model announcements omit environmental costs entirely.

We argue that **standardized carbon footprint reporting should become a routine component of ML research publications**. Without energy and emissions metrics, efficiency claims lack scientific rigor.

This is not environmental activism but scientific completeness. Physicists report energy in collision experiments;

---

[1]National Taiwan University, Taipei, Taiwan. Correspondence to: Guan-Ming Chiu <gmchiu@arbor.ntu.edu.tw>.

*Proceedings of the $43^{rd}$ International Conference on Machine Learning*, Seoul, South Korea. PMLR 306, 2026. Copyright 2026 by the author(s).

---

[1]Data sources: BERT (Strubell et al., 2019); T5, Meena, GShard-600B, GPT-3 (Patterson et al., 2021); OPT-175B (Zhang et al., 2022); BLOOM, Gopher (Luccioni et al., 2023); Llama-2 (Touvron et al., 2023); Llama-3.1 (Meta AI, 2024).

chemists report temperature in synthesis protocols. ML researchers claiming efficient methods should report energy consumption. Our position rests on four observations: (1) efficiency claims require specifying efficient *at what* (FLOPs, energy, carbon, or dollars); (2) reproducibility demands environmental context, as identical experiments in different locations yield vastly different carbon footprints; (3) the field lacks baselines to track sustainability over time; and (4) awareness plausibly drives change—while direct empirical evidence in ML is limited, analogous norms in other fields (code release, reproducibility checklists) suggest that what gets measured gets improved.

Now is the time to act. Measurement tools exist, researcher awareness is growing, and prominent releases including BLOOM, OPT, and Llama have already included carbon data, proving such reporting is feasible.

## 2. Related Work

Green computing research has studied data center efficiency for decades (Masanet et al., 2020). Within ML, Schwartz et al. introduced "Green AI" (Schwartz et al., 2020), and Strubell et al. brought attention to NLP's carbon costs (Strubell et al., 2019). Patterson et al. provided refined estimates for large-scale training (Patterson et al., 2021), while Luccioni et al. documented BLOOM's carbon footprint (Luccioni et al., 2023). Lottick et al. framed energy reporting as algorithmic accountability (Lottick et al., 2019), and recent work extends carbon analysis to adversarial ML and security domains (Hasan et al., 2024). K.C. et al. demonstrate that per-experiment carbon tracking is feasible beyond NLP and vision, integrating real-time CodeCarbon monitoring into cybersecurity anomaly detection workflows (K.C. et al., 2025). Henderson et al. found that the vast majority of papers at major venues omit energy data entirely (Henderson et al., 2020).

Our proposal draws on precedents from other fields: climate science frameworks for reporting uncertainty, medical research's CONSORT standards,[2] software engineering norms for code sharing, and prior work on structured reporting in ML (Dodge et al., 2019). Other scientific disciplines have begun addressing research carbon footprints, including astronomy (Knödlseder et al., 2022).

The scaling laws literature provides context for carbon costs. Kaplan et al. established performance power laws with respect to compute (Kaplan et al., 2020), and Hoffmann et al. showed that training smaller models on more data can achieve equivalent performance at lower cost (Hoffmann et al., 2022). Recent work has expanded understanding

of ML's footprint: MLPerf Power[3] established standardized efficiency measurement methodology (Tschand et al., 2025); studies show inference energy can exceed training costs for deployed models (Jegham et al., 2025; Fernandez et al., 2025); analyses project substantial increases in AI electricity demand (Xiao et al., 2025); and green AI techniques demonstrate 40–60% energy reductions without performance degradation (Verdecchia et al., 2023).

Several recent efforts have developed carbon-aware ML systems: OpenCarbonEval provides unified emission estimation (Yu et al., 2024); Clover enables carbon-aware inference routing (Li et al., 2023a); and CAFE addresses carbon-aware federated learning across distributed data centers (Bian et al., 2024). We argue these technical advances should be complemented by venue-level reporting standards, and offer concrete metrics, templates, and a phased adoption timeline toward this goal.

## 3. The Current State of Carbon Reporting in ML

Despite growing awareness of computational sustainability, carbon footprint reporting remains rare in ML publications. Henderson et al. conducted a systematic survey of 100 randomly sampled NeurIPS 2019 papers and found striking results: zero papers reported carbon impacts, only 1% reported any energy metrics, and just 17% reported compute-related metrics such as GPU-hours (Henderson et al., 2020). While awareness has grown since 2019, reporting remains the exception rather than the rule.

### 3.1. What Gets Reported

The standard computational details section of an ML paper typically includes hardware specifications such as GPU model, memory, and number of devices; training duration in wall-clock time or epochs; hyperparameters and optimization details; and dataset sizes with preprocessing steps. These details serve reproducibility but say nothing about environmental impact. A researcher reading that a model trained for 72 hours on 8 A100 GPUs cannot determine whether the experiment emitted 50 kg or 500 kg of $CO_2$ without additional information about data center location and energy sources.

### 3.2. Why Reporting Matters for Science

The absence of energy metrics creates several problems for scientific practice:

**Incomplete efficiency claims.** When a paper claims a model is "more efficient," this typically means fewer FLOPs

---

[2]Consolidated Standards of Reporting Trials, a guideline for reporting randomized controlled trials in medicine.

[3]An industry benchmark suite for measuring ML system performance and energy efficiency.

or faster inference. But FLOPs do not map linearly to energy consumption. Memory access patterns, hardware utilization, and batch sizes all affect energy use independently of operation counts. A model with 20% fewer FLOPs might consume the same energy if it has worse memory locality. Gowda et al. (2023) directly demonstrate that reducing FLOPs or parameters does not reliably yield proportional energy savings, because energy is governed by hardware-dependent factors such as memory access and GPU utilization; Desislavov et al. (2023), analyzing 94 ImageNet models, similarly find that the relationship between operations and energy is mediated by hardware efficiency rather than a simple linear mapping. Software-based power meters can themselves disagree with hardware meters by margins of comparable size (Jay et al., 2023), so moderate FLOP reductions can be entirely absorbed by non-compute costs and measurement variance.

**Non-reproducible comparisons.** Two research groups comparing methods on identical hardware may get different energy results depending on their locations. A GPU cluster in Quebec (hydroelectric grid, approximately 20 $gCO_2$/kWh) produces roughly 25× less carbon than one in Poland (coal-heavy grid, approximately 500 $gCO_2$/kWh) for the same computation (Lacoste et al., 2019). Without location data, carbon comparisons across papers become uninterpretable. Energy comparisons (kWh) and algorithmic efficiency metrics (FLOPs per token) remain valid without location data, but carbon footprint claims require this context.

**Hidden costs of progress.** When we celebrate a new state-of-the-art result, we rarely ask what it cost to achieve. The computational experiments behind a single paper can range from tens of GPU-hours to millions. Without systematic reporting, we cannot assess whether marginal accuracy gains justify orders-of-magnitude increases in compute.

**Lack of optimization incentives.** When energy costs are invisible, researchers have no incentive to optimize for them. A training run that uses 2× more energy than necessary due to suboptimal batch sizes or learning rate schedules will produce the same paper as an efficient run. If energy were reported, reviewers and readers could recognize and appreciate efficient implementations, creating positive incentives for carbon-conscious research practices.

### 3.3. A Worked Example

Consider two research groups training identical models with 4 V100 GPUs for 24 hours. Group A operates in Norway (29 $gCO_2$/kWh, PUE 1.1); Group B in Australia (650 $gCO_2$/kWh, PUE 1.4). Both consume ∼24 kWh of GPU energy (4 GPUs × 24 h × 0.25 kW) with identical GPU-hours, yet PUE and local grid intensity widen emissions ∼28×: ∼0.8 vs ∼22 $kgCO_2$eq. Without location data, these experiments appear equivalent on paper, and a reader cannot

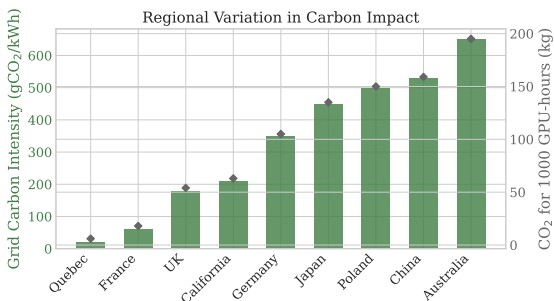

*Figure 2.* Regional variation in grid carbon intensity and resulting emissions for 1000 GPU-hours, ranging from 6 kg in Quebec to 195 kg in Australia.

tell whether "efficiency" came from algorithmic gains or a low-carbon grid; Figure 2 visualizes how this gap widens.

### 3.4. Quantifying the Costs

To make the economic and environmental stakes concrete, we summarize publicly disclosed costs for representative systems (full breakdown in Table 7, Appendix C). Training GPT-3 (Patterson et al., 2021) cost roughly $154,000 in electricity at average US industrial rates; Llama-3.1 405B (Dubey et al., 2024) emitted carbon comparable to the annual emissions of about 2,200 average US households; and at deployment scale a model serving 10M queries per day at ∼0.003 kWh/query (representative of large generative model inference (Luccioni et al., 2024; Samsi et al., 2023)) consumes on the order of 10,950 MWh per year, which can exceed training within months. Set against the survey above, where 91.0% of papers report no energy data, costs of this magnitude remain effectively invisible across the vast majority of published research.

### 3.5. How Often Do Papers Report Carbon? An ICML 2025 Survey

To quantify how the gap has evolved since 2019, we surveyed 1,000 randomly sampled ICML 2025 accepted papers (from 3,260 total) using automated keyword matching with manual verification across three categories: compute metadata, energy data, and carbon data. Analysis code and per-paper results are released alongside this paper. Figure 3 contrasts our results with the NeurIPS 2019 baseline of Henderson et al. (2020).

Compute metadata reporting has grown substantially (17% to 57.3%), plausibly reflecting reproducibility checklists and venue-level expectations. Yet energy and carbon reporting remain in single digits despite high-profile disclosures from BLOOM, OPT, and Llama. Energy (9.0%) and carbon (5.9%) reporting today resemble compute reporting circa

2019: this is the awareness-action gap our paper identifies, and our proposal aims to catalyze a similar trajectory through clear expectations, templates, and gradual normalization.

### 3.6. Existing Tools and Their Limitations

Several tools exist for measuring ML carbon footprints: CodeCarbon (CodeCarbon Contributors, 2020), ML CO2 Impact[4] (Lacoste et al., 2019), Carbontracker (Anthony et al., 2020), and experiment-impact-tracker (Henderson et al., 2020). These tools can estimate energy consumption and convert it to carbon emissions using regional grid intensity data.

However, tool availability has not led to widespread adoption, primarily because venues do not encourage or reward carbon reporting.

### 3.7. The Gap Between Awareness and Action

The ML community is aware of environmental concerns: workshops on climate change and AI regularly draw submissions, and papers on efficient methods often mention environmental benefits. Yet this awareness rarely translates into consistent reporting.

Several factors explain this gap. First, there is no template. Authors who want to report carbon footprint must decide what to measure, how to measure it, and how to present results. Without guidance, many default to reporting nothing. Second, there is no enforcement. Reviewers do not expect carbon data, so its absence is not penalized. Third, there is uncertainty about accuracy. Researchers worry that imprecise estimates will be criticized, so they prefer to omit data rather than report uncertain figures.

This pattern is not unique to environmental reporting. The

[4]ML CO2 Impact calculator: `https://mlco2.github.io/impact/`

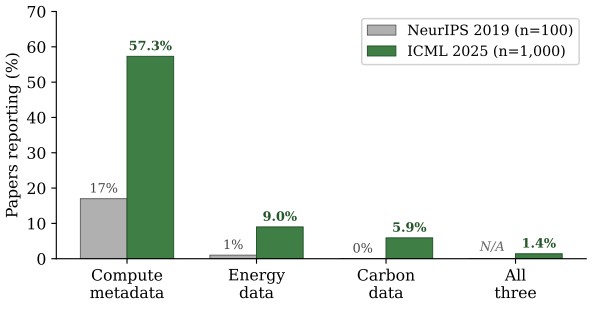

*Figure 3.* Reporting rates for compute, energy, and carbon metadata: NeurIPS 2019 (Henderson et al., 2020) versus our ICML 2025 survey. "All three" was not separately reported in 2019.

field has faced similar challenges with statistical practices (e.g., reporting confidence intervals), code release (now expected), and dataset documentation. In each case, progress required venues to establish clear expectations and provide templates that make compliance straightforward.

## 4. Proposed Reporting Guidelines

The gap between awareness and action described above will not close on its own. Without concrete standards, authors face uncertainty about what to report and how, leading most to report nothing. Standardization reduces this friction by providing clear templates and shared expectations.

We propose standardized reporting guidelines for carbon footprint in ML research. The goal is straightforward: every paper should report the energy consumed and carbon emitted by its experiments, using consistent metrics that enable comparison across studies. Our guidelines prioritize practicality over perfection—estimates with acknowledged uncertainty are preferable to no data at all.

### 4.1. Standardized Metrics for Reporting

The following metrics align with established carbon accounting standards: the Greenhouse Gas Protocol (World Resources Institute and World Business Council for Sustainable Development, 2004) for organizational carbon accounting[5] and the Software Carbon Intensity (SCI) specification (Green Software Foundation, 2022) for software applications.

Building on these frameworks, papers should report five key metrics where applicable:

1. **Total energy consumption (kWh):** Direct measurement via hardware power meters provides the most accurate values. When unavailable, software-based tools such as CodeCarbon, Carbontracker, or nvidia-smi can provide reasonable estimates, though researchers should acknowledge measurement uncertainty (Jay et al., 2023). For cloud workloads, providers increasingly offer energy consumption data through APIs and dashboards.

2. **Carbon emissions (kgCO$_2$eq):** Calculated by multiplying energy consumption by grid carbon intensity, following the methodology outlined in the GHG Protocol. The unit kgCO$_2$eq (kilograms of carbon dioxide equivalent) accounts for all greenhouse gases converted to their CO$_2$ warming potential.

3. **Grid carbon intensity (gCO$_2$/kWh):** This value varies dramatically by region, from approximately

[5]ISO 14064-1:2018 provides complementary international standards for emissions verification.

20 $gCO_2$/kWh in hydroelectric-dominated Quebec to over 700 $gCO_2$/kWh in coal-heavy grids. Researchers should cite their data source, such as regional grid operators, Electricity Maps[6] (Electricity Maps, 2023), or annual averages from governmental statistics. Time-of-use variations can also be significant.

4. **Power Usage Effectiveness (PUE):** This ratio captures data center overhead including cooling, lighting, and power distribution losses. A PUE of 1.0 represents perfect efficiency; typical values range from 1.1 for hyperscale cloud data centers to 1.5 or higher for older academic facilities. When unknown, a conservative estimate of 1.2–1.4 should be stated explicitly.

5. **Compute region:** The geographic location (country, state, or cloud availability zone) enables readers to verify or recalculate carbon estimates and facilitates meta-analyses across studies.

Following the GHG Protocol's emissions categorization, ML training typically falls under Scope 2 (purchased electricity), while Scope 3 includes embodied carbon from hardware manufacturing and supply chain emissions (Gupta et al., 2022). For comprehensive lifecycle assessment, researchers conducting large-scale training should consider reporting embodied carbon estimates, though we recognize this requires additional data that may not always be available (Ligozat et al., 2022).

For cloud experiments, Google Cloud, Microsoft Azure, and AWS all offer carbon footprint dashboards that researchers can cite directly.

**Water footprint as a supplementary metric.** Beyond electricity, data center cooling consumes substantial freshwater. Li et al. (2023b) estimate that training GPT-3 in Microsoft's US data centers consumed on the order of 700,000 liters of freshwater, with comparable off-site water tied to electricity generation. Because measurement tools for water are less mature than for energy and carbon, we recommend water footprint (in liters or $m^3$) as a *supplementary* metric for large-scale training. Where direct measurement is infeasible, authors can cite published water usage effectiveness (WUE) values for their data center or region.

### 4.2. Measurement Tools and Carbon-Aware Benchmarks

Adoption of carbon reporting depends heavily on reducing friction. Several open-source tools already exist for this purpose, including CodeCarbon (CodeCarbon Contributors, 2020), Carbontracker (Anthony et al., 2020), and experiment-impact-tracker (Henderson et al., 2020).

Software-based estimates generally track hardware measurements within 20–30% for GPU-intensive workloads, which is sufficient for the transparency goals of carbon reporting. Even with this uncertainty, cross-paper comparisons remain meaningful: a $10\times$ difference between methods is clearly distinguishable from measurement noise, and trend analysis across hundreds of papers will average out individual errors.

Ideally, measurement should require minimal code changes. Decorators or context managers make tracking nearly invisible:

```
@track_emissions(project="training")
def train():
    model.fit(X, y, epochs=100)
```

Output formats should be standardized across tools, and HPC clusters should enable job-level energy accounting through schedulers like SLURM.[7]

Beyond paper reporting, community benchmarks should incorporate carbon metrics. MLPerf Power has established methodology for standardized efficiency benchmarking (Tschand et al., 2025). Leaderboards could display carbon cost per accuracy point, enabling identification of Pareto-optimal models. Green AI techniques can reduce energy consumption substantially (Verdecchia et al., 2023), but these gains remain invisible without carbon-aware evaluation.

### 4.3. Choice of Reporting Primitives

Different reporting units serve different purposes, and the community must choose which primitives to standardize. We consider several candidates and their tradeoffs:

**Energy-based primitives.** Metrics like kWh (total energy) or Wh/1000 tokens (energy intensity) measure physical resource consumption independent of location. These are reproducible and enable fair comparison of algorithmic efficiency across institutions. However, they do not capture environmental impact, which depends on carbon intensity.

**Carbon-based primitives.** Metrics like $kgCO_2eq$ (total emissions) or $gCO_2$/1000 tokens directly measure environmental impact. These are the ultimate quantity of interest for sustainability but depend on location and infrastructure, making cross-study comparison difficult without normalization.

**Compute-normalized metrics.** Metrics like kWh/billion parameters or $gCO_2$/accuracy-point enable comparison across models of different scales. These are useful for identifying efficient architectures but require careful definition of the denominator.

---

[6] https://electricitymaps.com

[7] Simple Linux Utility for Resource Management, a widely used workload manager for high-performance computing clusters.

**System vs. paper-level reporting.** Benchmarks like MLPerf Power (Tschand et al., 2025) measure system efficiency under controlled conditions (fixed hardware, standardized workloads, comparable environments). Paper-level reporting captures the full cost of research including failed experiments and hyperparameter search. Both are valuable: system benchmarks enable hardware comparison while paper-level reporting provides scientific transparency.

We recommend a layered approach: papers should report *both* energy (kWh) and carbon (kgCO$_2$eq), along with the inputs needed to verify the calculation (grid intensity, PUE, region). This enables readers to compare energy efficiency directly while understanding the carbon implications. For inference, standardized benchmarks (e.g., Wh per 1000 tokens at batch size 32) complement paper-specific totals.

### 4.4. Reporting Training and Inference Separately

Training and inference have fundamentally different carbon characteristics and must be reported separately (Luccioni et al., 2024; Samsi et al., 2023; Wu et al., 2022; Jegham et al., 2025).

**Training carbon** is a one-time, upfront investment. It scales with model size following established scaling laws: larger models require more compute, and compute requirements grow polynomially with parameter count (Kaplan et al., 2020; Hoffmann et al., 2022). The Chinchilla scaling laws suggest that compute-optimal training involves more tokens rather than simply larger models (Hoffmann et al., 2022). Training can be scheduled strategically to exploit periods of low grid carbon intensity or high renewable availability (Dodge et al., 2022), and recent work on multi-day carbon intensity forecasting enables predictive rather than reactive scheduling (Maji et al., 2023). Once complete, training carbon can be amortized across all downstream applications.

**Inference carbon** accumulates continuously with each query served. Recent studies demonstrate that inference energy follows different scaling relationships than training (Desislavov et al., 2023; Samsi et al., 2023; Fernandez et al., 2025). Energy per token decreases with batch size but increases with sequence length. A large language model can consume several kWh per 1000 queries depending on model size, hardware, and configuration (Luccioni et al., 2024). For frontier models like GPT-3 (Brown et al., 2020) or Llama (Touvron et al., 2023) deployed at massive scale, inference emissions can exceed training emissions within weeks or months (Chien et al., 2023).

This distinction has critical implications. Research papers typically report only training costs, but commercially deployed models incur ongoing inference costs that dwarf training investments. Papers should report training carbon with methodology details, and model releases should include standardized inference energy benchmarks to enable lifecycle carbon assessment.

Inference benchmarks must be domain-specific. For example, language models might report energy per 1000 tokens at specified batch sizes, while vision models might use energy per 1000 images. Classical ML, recommendation systems (Wegmeth et al., 2025), and AutoML have different profiles where per-sample metrics may be more appropriate. The key principle is standardization within each domain to enable fair comparison, while acknowledging that production deployments involve additional complexity that simplified benchmarks do not fully capture.

### 4.5. Measurement Protocol

To enable meaningful comparisons across papers, we propose a minimal viable protocol that specifies not just which metrics to report, but how to measure them. This protocol prioritizes comparability and transparency while acknowledging practical constraints.

**How to measure.** Report average power draw during the measurement period, not peak power or TDP, as nameplate ratings can overestimate by 20–40%. When using software-based tools, specify the sampling interval (e.g., 1-second intervals for CodeCarbon); for hardware power meters, specify the measurement point (GPU only, server, or rack-level). Software-based tools have known limitations, so report confidence intervals rather than false precision (e.g., "approximately 150 kWh $\pm 20\%$" rather than "153.7 kWh"). When possible, cross-validate estimates using multiple tools.

**What to include.** Clearly define what computation is included. The "final reported experiments" boundary should include all training runs whose results appear in the paper, final evaluation runs, and any experiment-specific preprocessing. Pre-trained weights need not be counted, but authors should cite reported training emissions of base models when available. When results are averaged across $n$ runs, report total energy ($n\times$ single-run), not per-run averages. Report hyperparameter search separately from final training: "Final training: X kWh; Hyperparameter search: Y kWh (Z configurations evaluated)."

**How to report.** For multi-location compute, report contributions separately with location-specific carbon intensities, then provide a total (e.g., "Location A (Quebec): 500 kWh, 10 kgCO$_2$eq; Location B (Germany): 300 kWh, 105 kgCO$_2$eq; Total: 800 kWh, 115 kgCO$_2$eq"). Include normalized metrics (carbon per accuracy point, per training sample, or per parameter) alongside totals to enable fair comparison across different scales.

# 5. Call to Action

We call on the ML community to take concrete steps toward normalizing carbon footprint reporting.

## 5.1. Venues and Reviewers

Conference organizers hold the key to widespread adoption. We urge major venues (NeurIPS, ICML, ICLR) to add optional carbon reporting fields to submission forms within the next cycle (hardware, location, energy, carbon), commission best-practice guides, transition from optional to encouraged across subsequent stages, and publish annual aggregate statistics so the field can track its trajectory. Reviewers must understand that carbon reporting serves transparency, not gatekeeping: papers should never be penalized for reporting high carbon costs (which would create perverse incentives to underreport). Scrutiny should focus on two cases: (1) efficiency claims without energy data, and (2) massive compute yielding marginal gains over simpler baselines. High carbon is acceptable when justified; the goal is informed evaluation, not carbon-based rejection.

## 5.2. Researchers and Institutions

Authors need not wait for venue requirements. Voluntary adoption builds community norms. When reporting, authors should document measurement methodology clearly, including tool versions, hardware specifications, and data center locations. Research teams can begin by integrating measurement tools into their training pipelines and establishing lab-level reporting practices that normalize carbon awareness within their groups.

Universities, national labs, and cloud providers can facilitate reporting by providing energy monitoring infrastructure and publishing PUE ratios and energy sources. Some institutions have begun offering carbon-aware scheduling, routing jobs to times when grid carbon intensity is lower. Institutional support is particularly important for researchers who lack direct access to power monitoring hardware; centralized infrastructure can provide the data that individual researchers cannot easily obtain.

**The incentive gap.** We should be candid about a structural disanalogy: code release confers direct reciprocal benefits (citations, downstream use, visibility) that carbon reporting does not. No extra citations accrue to reporting kWh, and intrinsic motivation alone will plateau adoption below code-release levels. Three mechanisms can partially close the gap: *venue-level expectation* (as with the NeurIPS reproducibility checklist, which reached widespread adoption in two years (Pineau et al., 2021)); *external pulls* from universities and funding agencies that increasingly require sustainability reporting, making paper-level carbon data directly reusable for

institutional audits; and *carbon-aware leaderboards* (Section 4.2) that create a competitive signal accruing to authors. Acknowledging this asymmetry honestly is more useful than overstating the analogy: the proposal must work *despite* the weaker incentive structure.

## 5.3. Preventing Gaming and Strategic Underreporting

Any reporting requirement creates incentives for strategic behavior. We identify two categories of potential gaming and propose mitigations.

**Selective reporting scope.** Authors might cherry-pick which runs to include, reporting energy only for the best-performing run while averaging accuracy across multiple runs. Similarly, authors might define experiment boundaries to exclude expensive hyperparameter sweeps or omit preprocessing, data loading, and checkpointing from measurements. The measurement protocol addresses these concerns by requiring total energy across all averaged runs, separate reporting of search and training compute, and explicit documentation of what is excluded and why.

**Carbon accounting manipulation.** Authors using renewable energy credits (RECs)[8] or carbon offsets might report only net emissions, obscuring gross energy consumption. Following GHG Protocol guidance, we require reporting of both gross and net emissions, with energy consumption in kWh reported independently. A related concern is location arbitrage, where authors route computation through low-carbon regions primarily for reporting optics. However, this is actually a positive outcome: if reporting incentivizes using cleaner grids, that represents genuine emissions reduction.

We acknowledge that no reporting system is gaming-proof, but establishing clear norms makes egregious underreporting socially costly and statistically detectable.

## 5.4. Timeline for Adoption

We propose a phased transition in three stages, modeled on successful precedents: the shift from optional to expected code release, and the rapid adoption of NeurIPS reproducibility checklists (Pineau et al., 2021). The timeline is deliberately gradual: each stage is designed to be *minimal in itself* while producing the data and infrastructure that make the next stage realistic. The Stage 1 ask is intentionally tiny, only what 57.3% of papers already partially provide (Section 3.5); Stage 2 builds on the aggregate carbon estimates that Stage 1 enables venues to compute; Stage 3 then applies full carbon accounting only to large-scale experiments, leveraging the empirical ranges established in Stage 2. This scaffolding is what distinguishes our proposal

---

[8]Tradable certificates representing proof that electricity was generated from renewable sources.

from a sudden mandate.

**Stage 1: Foundation.** Major venues add optional carbon reporting fields to submission forms, asking for GPU-hours, hardware specifications (GPU model, count), and data center location if known. Reviewers check only that reported data is present and internally consistent—for example, 8 A100 GPUs for 24 hours at typical utilization should yield roughly 50–150 kWh depending on hardware variant and workload, and gross inconsistencies (e.g., order-of-magnitude errors) warrant clarification but not rejection. Venues host workshops and tutorials on measurement tools, and early adopters share case studies documenting their reporting experience.

**Stage 2: Encouragement.** Venues publish aggregate statistics from submitted papers, establishing empirical ranges for different experiment types (e.g., fine-tuning a 7B model typically consumes 50–200 kWh; training a vision transformer from scratch typically consumes 500–2000 kWh). The expected requirement expands to measured energy (kWh) with acknowledged uncertainty bounds. Community benchmarks such as MLPerf incorporate efficiency metrics alongside accuracy. Third-party organizations including the Green Software Foundation offer voluntary verification services. Cross-venue data sharing agreements enable longitudinal analysis, and institutions begin offering carbon-aware job scheduling.

**Stage 3: Normalization.** Reporting becomes a community expectation rather than an exception. For large-scale experiments (exceeding thresholds such as 1000 GPU-hours or 10,000 kWh), full reporting of all five metrics becomes standard: kWh, kgCO$_2$eq, grid intensity, region, and PUE. Independent auditing frameworks emerge for frontier model training, potentially aligned with ISO 14064 certification standards. Venues recognize verified reports with badges similar to artifact evaluation, and carbon-aware practices integrate into graduate curricula.

Success at each stage is measured by concrete metrics: (1) *Adoption rate*—targeting 30% of papers including basic data (Stage 1), 50% including measured energy (Stage 2), and 70% meeting full requirements for applicable papers (Stage 3); (2) *Data quality*—fraction of reports that are internally consistent and include uncertainty estimates; (3) *Trend visibility*—ability to answer questions like "Is NLP research becoming more carbon-efficient?" with statistical confidence.

### 5.5. Privacy and Double-Blind Review

Reporting grid carbon intensity, PUE, or compute region can substantially narrow author identity, conflicting with double-blind review. A rare data center, a national grid, or even a cloud availability zone can be enough to deanonymize a research group. We treat this as a first-order design constraint

and propose two complementary mitigations. **Tiered visibility:** carbon metadata that exposes location is submitted as supplementary information visible only to area and program chairs, while reviewers see the de-identified portion (GPU hardware, energy in kWh, normalized carbon), which suffices for plausibility checks. **Reference-grid normalization during review:** authors report carbon normalized to a fixed reference grid (e.g., 475 gCO$_2$/kWh, the global average), revealing no geographic information; the actual region and grid intensity are disclosed only in the camera-ready. Venues can adopt either or combine both. Carbon transparency at publication and anonymity at review are compatible if the data flow is designed for both.

## 6. Alternative Views

A proposal of this scope raises legitimate concerns. We address four common objections.

### 6.1. Measurement Is Too Difficult

Critics argue accurate energy measurement requires specialized hardware or configurations many researchers lack, and different tools yield different numbers. We acknowledge this: software tools vary by 20–30% from ground truth (Jay et al., 2023) and exhibit systematic biases (CodeCarbon tends to underestimate relative to hardware meters), so we recommend reporting the tool and version used to enable cross-calibration. The barrier to entry is lower than commonly perceived: CodeCarbon requires only a pip install, major cloud providers offer built-in dashboards, and HPC clusters increasingly log energy at the job level via SLURM. We advocate tiered requirements: at minimum GPU-hours and hardware, ideally measured kWh, with full carbon accounting where feasible. Imperfect measurement is better than no measurement; astronomy and climate science routinely report uncertainty ranges, and even with 20–30% error margins, energy data reveals order-of-magnitude differences and enables trend analysis.

### 6.2. Reporting Creates Barriers for Under-Resourced Groups

A more serious concern is equity. Researchers at well-funded institutions with efficient data centers will report lower carbon footprints than those at smaller institutions using older hardware or dirtier grids, which could disadvantage already-marginalized researchers. We acknowledge a tension in our proposal: while we recommend against using carbon as a review criterion (Section 5.1), we also suggest that benchmarks incorporate carbon metrics (Section 4.2), creating a foreseeable "carbon leaderboard" effect where high-carbon regions face reputational disadvantage even without formal acceptance penalties. We propose three mechanisms:

**Dual reporting of energy and carbon.** Leaderboards should display both kWh (algorithmic/hardware efficiency, independent of location) and kgCO$_2$eq (full environmental impact). High-carbon regions can demonstrate strong energy efficiency even if carbon numbers are higher, distinguishing geographic constraints from engineering choices.

**Reference-grid normalization.** For comparative rankings, venues could report carbon emissions normalized to a standard reference grid intensity (e.g., the global average of approximately 475 gCO$_2$/kWh). This "what-if" metric answers: "What would this experiment emit on an average grid?" Actual emissions should still be reported for transparency, but normalized values enable fairer cross-institution comparison of algorithmic efficiency. *Dual reporting in practice.* Normalization can mislead in isolation: consider Experiment A in Quebec (1,000 kWh, 20 gCO$_2$/kWh; actual 20 kgCO$_2$eq, normalized 475 kgCO$_2$eq) versus Experiment B in Germany (400 kWh, 350 gCO$_2$/kWh; actual 140 kgCO$_2$eq, normalized 190 kgCO$_2$eq). Looking only at actual carbon, A appears far "cleaner"; looking only at normalized, B appears more efficient. Both readings are partial; only the dual view shows that A is carbon-friendly today because of grid composition while B uses 60% less energy and is the algorithmically more efficient method. Leaderboards displaying both numbers side-by-side prevent either misinterpretation.

**Efficiency ratios over absolute values.** Metrics like kWh per accuracy point or gCO$_2$eq per billion parameters reward efficiency regardless of scale, recognizing scientific craftsmanship that absolute totals would obscure.

For researchers in the Global South or at institutions with minimal infrastructure support, we recommend practical accommodations: software-based tools require only pip install, and approximate reporting (e.g., GPU-hours with estimated regional intensity) should be accepted when precise measurement is infeasible. The goal is inclusion, not exclusion. Reporting must never be a gatekeeping criterion: a breakthrough remains valuable regardless of its carbon cost; what matters is that the cost is known, so the community can weigh trade-offs explicitly rather than ignoring them.

### 6.3. Standards Are Premature

Some argue the field lacks consensus on measurement methods, and standardizing now will lock in imperfect approaches. We disagree: perfect standards are not required for useful reporting. The current state, where most papers report nothing, prevents progress on understanding the field's environmental trajectory, and even imperfect data enables trend analysis. Moreover, standardization drives improvement: once venues encourage reporting, tool developers gain stronger incentives to improve accuracy and the community converges on best practices through experience.

### 6.4. Focus Should Be on Industry, Not Academia

A fourth objection holds that academic research contributes a small fraction of total ML compute. The real environmental impact comes from industry training runs and deployed inference at scale. Requiring academics to report while industry operates opaquely is asymmetric and ineffective.

We agree that industry must be part of the solution, but academic norms influence industry practice: researchers who learn to report environmental impact in graduate school carry these practices into industry roles, and academic venues can require that industry-sponsored publications meet the same standards, creating pressure for organizational transparency. As venues publish aggregate statistics, industry papers that report nothing become conspicuous outliers. The objection also underestimates academic impact: aggregate compute across thousands of papers is substantial, and academic research shapes what problems the field considers important; if efficiency and environmental impact become standard metrics in academic publications, industry will face pressure to adopt similar standards.

## 7. Conclusion

Carbon footprint reporting is the logical next step in the ML community's evolving reporting standards: efficiency claims without energy data are incomplete, analogous to reporting accuracy without specifying the test set. Systematic reporting would let us establish baselines, track progress, and answer questions we currently cannot (Is ML becoming more carbon-efficient? Do algorithmic gains translate to real energy savings?). The field has adopted comparably difficult norms before, including code release, NeurIPS reproducibility checklists (Pineau et al., 2021), and Datasheets for Datasets (Gebru et al., 2021), with transparency benefits outweighing costs. Equity must be designed in: dual energy-carbon reporting, reference-grid normalization, and approximate estimates make reporting inclusive rather than exclusionary. **Future work** extends this training-side proposal to inference, whose accumulating, deployment-dependent costs require domain-specific benchmarks (energy per 1000 tokens for LLMs, per 1000 images for vision, per sample for classical ML). The path forward requires coordination across venues, authors, and institutions; the tools and precedents exist, and this norm is overdue.

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

# A. Glossary of Key Terms

*Table 1.* Glossary of carbon footprint reporting terminology.

| Term | Definition |
| --- | --- |
| **Carbon Footprint** | The total greenhouse gas emissions caused directly and indirectly by an activity, expressed as carbon dioxide equivalent ($CO_2$eq). |
| **Carbon Intensity** | The amount of $CO_2$ emitted per unit of electricity generated, typically measured in $gCO_2$/kWh. Varies by region and time based on energy mix. |
| **$CO_2$eq** | Carbon dioxide equivalent; a standard unit for measuring carbon footprints that converts all greenhouse gases to the equivalent amount of $CO_2$ based on global warming potential. |
| **GPU-hours** | A measure of computational work equal to one GPU operating for one hour. Does not account for GPU utilization or power draw variations. |
| **PUE** | Power Usage Effectiveness; the ratio of total facility energy to IT equipment energy. A PUE of 1.0 means all power goes to computing; typical data centers range from 1.1 to 2.0. |
| **Scope 1 Emissions** | Direct emissions from owned or controlled sources (e.g., on-site generators). |
| **Scope 2 Emissions** | Indirect emissions from purchased electricity, steam, heating, and cooling. Most ML carbon footprints fall here. |
| **Scope 3 Emissions** | All other indirect emissions in the value chain, including hardware manufacturing and end-of-life disposal. |
| **TDP** | Thermal Design Power; the maximum amount of heat a component is designed to dissipate, often used as a proxy for power consumption. |
| **FLOPs** | Floating-point operations; a measure of computational work independent of hardware or energy consumption. |
| **Embodied Carbon** | The carbon emissions associated with manufacturing, transporting, and disposing of hardware, distinct from operational emissions. |

# B. Sample Carbon Reporting Templates

The three templates below correspond to the phased adoption stages described in Section 5.4. Each template specifies only the minimum required fields, but authors are encouraged to disclose additional context where available (e.g., job-level energy traces, intra-run grid intensity variation, or hardware utilization timelines) since such metadata accumulates value for community-level meta-analyses even when not strictly required at submission time.

The templates follow a principle of *progressive disclosure*: Stage 1 asks only for what most authors already know (hardware and runtime), Stage 2 adds a measured energy figure once tooling is in place, and Stage 3 collects the full set of variables needed to verify a carbon estimate from first principles. Authors at any stage may opt into a higher tier voluntarily, and we expect mixed adoption during the transition period. Reviewers should treat these templates as descriptive scaffolds rather than rigid checklists: a paper that reports four of the five Stage 3 fields with a clearly stated reason for the omission is more useful than one that reports five fields with no methodology trail.

When values are estimated rather than measured, authors should mark them explicitly (e.g., "estimated from TDP") and report the assumption used. A consistent convention across the community, even at the cost of some precision, is more valuable than a patchwork of incompatible accounting choices.

*Table 2.* Minimal reporting (Stage 1): GPU-hours and hardware only.

| Metric | Value |
| --- | --- |
| Hardware | 2× NVIDIA RTX 3090 |
| Total GPU-hours | 48 |
| Compute region | University cluster (unknown grid) |
| *Energy/carbon not measured; estimated ∼15 kWh based on TDP.* | |

*Table 3.* Standard reporting (Stage 2): includes measured energy. *Illustrative example; values are hypothetical.*

| Metric | Value |
| --- | --- |
| Hardware | 8× NVIDIA A100 (40GB) |
| Total GPU-hours | 2,400 |
| Compute region | US-West (California) |
| Grid carbon intensity | 210 $gCO_2$/kWh |
| Data center PUE | 1.1 |
| Total energy (estimated) | 1,000 kWh |
| Total carbon (estimated) | 210 $kgCO_2$eq |
| Measurement tool | CodeCarbon v2.1 |

*Table 4.* Comprehensive reporting (Stage 3): full carbon accounting for large-scale experiments. *Illustrative example; values are hypothetical.*

| Category | Details |
| --- | --- |
| *Hardware Configuration* | |
| GPU Type | NVIDIA H100 (80GB) |
| Number of GPUs | 256 |
| CPU | AMD EPYC 7763 (per node) |
| Interconnect | InfiniBand NDR 400Gb/s |
| *Compute Summary* | |
| Total GPU-hours | 86,016 (256 GPUs × 14 days) |
| Average GPU Utilization | 78% |
| Peak Power Draw (measured) | 148 kW |
| *Location & Infrastructure* | |
| Data Center Location | Iowa, USA |
| Grid Carbon Intensity | 380 $gCO_2$/kWh (annual average) |
| Renewable Energy Credits | 50% offset claimed |
| PUE | 1.08 |
| *Energy & Emissions* | |
| Total Energy (measured) | 62,000 kWh |
| Scope 2 Emissions (gross) | 23,560 $kgCO_2$eq |
| Scope 2 Emissions (net, after RECs) | 11,780 $kgCO_2$eq |
| *Contextual Comparisons* | |
| Equivalent car miles | 59,000 miles |
| Equivalent transatlantic flights | 13 round trips |
| Measurement Method | Direct power metering + CodeCarbon validation |

# C. Reference Data

*Table 5.* Grid carbon intensities by region (recent averages). Sources: Electricity Maps, IEA Emissions Factors 2025, Ember Global Electricity Review 2025, Carbon Brief.

| Region | gCO$_2$/kWh | Cloud Zone |
|---|---|---|
| Quebec | 20 | GCP: na-northeast1 |
| Norway | 29 | Azure: Norway East |
| France | 60 | GCP: europe-west9 |
| UK | 125 | Azure: UK South |
| California | 210 | GCP: us-west1 |
| Massachusetts | 280 | AWS: us-east-1 |
| Germany | 350 | AWS: eu-central-1 |
| US (average) | 384 | — |
| Global (average) | 475 | — |
| Poland | 500 | — |
| China | 560 | Alibaba: cn-hangzhou |
| Australia | 650 | GCP: au-southeast1 |
| India | 700 | AWS: ap-south-1 |

*Table 6.* Typical GPU power consumption during training. TDP from NVIDIA/AMD specifications; typical training power at 80–85% TDP.

| GPU | TDP | Typical Training |
|---|---|---|
| A10 | 150W | 120W |
| V100 (32GB) | 300W | 250W |
| L40 | 300W | 250W |
| RTX 3090 | 350W | 290W |
| A100 (40GB) | 400W | 330W |
| A100 (80GB) | 400W | 350W |
| RTX 4090 | 450W | 380W |
| MI250X | 560W | 470W |
| H100 (80GB) | 700W | 580W |

*Table 7.* Representative training and inference costs that current ML papers leave unreported. Assumptions: US industrial electricity $0.12/kWh; inference per-query energy from (Luccioni et al., 2024; Samsi et al., 2023).

| System | Energy (MWh) | Carbon (tCO$_2$eq) | Equiv. |
|---|---|---|---|
| *Training (one-time)* | | | |
| GPT-3 (Patterson et al., 2021) | 1,287 | 552 | $154K |
| Llama-3.1 405B (Dubey et al., 2024) | — | 8,930 | 2,200 households |
| *Inference (annual, 10M queries/day)* | | | |
| 0.003 kWh/query (Luccioni et al., 2024; Samsi et al., 2023) | 10,950 | varies | training-scale |

**Formulas.** Total energy = $E_{\text{compute}} \times$ PUE; carbon = $E_{\text{total}} \times I$ (grid intensity); energy estimate = GPU-hours $\times P_{\text{GPU}} \times U$ (GPU at 70–80% TDP, plus 10–20% overhead).

**Usage.** The three tables compose multiplicatively. Table 6 provides $P_{\text{GPU}}$, multiplying by GPU-hours yields compute energy, applying PUE gives total energy, and Table 5 converts energy to carbon by region. Authors lacking direct meter access can still produce a Stage-1 estimate from hardware specifications alone, and Table 7 provides scale anchors for situating the resulting numbers against published frontier-model training and inference costs.

## D. Existing Tools and Standards

*Table 8.* Overview of carbon footprint measurement tools for ML research.

| Tool | Measurement Method | Features & Limitations |
|---|---|---|
| CodeCarbon | Software-based power estimation using Intel RAPL[a] and nvidia-smi[b] | Easy integration with Python; cross-platform; may underestimate by 10–20% compared to hardware meters |
| Carbontracker | Software-based with hardware abstraction | Epoch-level tracking; predictive estimates; limited to Linux |
| experiment-impact-tracker | Software-based with regional carbon data | Comprehensive logging; JSON output; academic-focused |
| ML CO2 Impact | Web calculator using GPU-hours | Quick estimates; no code integration; uses average power values |
| Cloud Carbon Footprint | Cloud provider APIs | Works with AWS, GCP, Azure; relies on provider-reported data |
| *Hardware-based Methods* | | |
| Power meters (e.g., Watts Up Pro) | Direct electrical measurement | Most accurate; requires physical access; measures at outlet level |
| IPMI/BMC sensors[c] | Server-level power reporting | Built into enterprise servers; 5–10% accuracy |
| PDU metering[d] | Rack-level power monitoring | Available in data centers; includes all equipment in rack |

[a]Running Average Power Limit, Intel's interface for energy consumption monitoring. [b]NVIDIA System Management Interface, a command-line tool for querying GPU power draw. [c]Intelligent Platform Management Interface / Baseboard Management Controller. [d]Power Distribution Unit.

*Table 9.* Relevant standards and frameworks for carbon reporting.

| Standard/Framework | Description & Relevance |
|---|---|
| GHG Protocol | The most widely used international standard for corporate carbon accounting. Defines Scope 1, 2, and 3 emissions categories. |
| ISO 14064 | International standard for quantification and reporting of greenhouse gas emissions. Provides verification requirements. |
| ISO 14067 | Standard specifically for carbon footprint of products, applicable to ML models as products. |
| Science Based Targets initiative (SBTi) | Framework for setting emission reduction targets aligned with climate science. Increasingly adopted by tech companies. |
| IEEE P2874 | Proposed standard for carbon footprint metrics specifically for AI systems (under development). |
| Green Software Foundation | Industry consortium developing standards for sustainable software, including the Software Carbon Intensity (SCI) specification. |

## E. Worked Example

A research team fine-tunes a 7B model: $4\times$ A100 GPUs, 48 hours, Massachusetts (280 $gCO_2$/kWh), PUE 1.4.

**Calculation:** GPU-hours = $4 \times 48 = 192$. GPU energy = $192 \times 0.33$ kW $\approx 63$ kWh. With 15% overhead and PUE: $63 \times 1.15 \times 1.4 \approx 102$ kWh. Carbon $\approx 102 \times 0.28 \approx 29$ $kgCO_2$eq ($\pm20\%$).

**Report:** $\approx29$ $kgCO_2$eq ($\pm20\%$) $\approx 70$ car miles. The same experiment in Quebec (20 $gCO_2$/kWh) would produce $\approx2$ kg; in Australia (650 $gCO_2$/kWh), $\approx66$ kg—a $33\times$ variation.

