# OpenReview forum: "Position: Carbon Footprint Reporting Should Be Routine in Machine Learning Research"
_ICML.cc/2026/Position_Paper_Track — ICML 2026 Position Paper Track regular_

### Official Review · Reviewer_z2pR · 2026-03-11

**Significance:** 4
**Argument Clarity:** 4
**Rating:** 5
**Confidence:** 4

**Questions:**

- The paper claims that "what gets measured gets improved," drawing an analogy to code release and reproducibility checklists. However, code release and reproducibility norms directly benefit the researchers who adopt them by enabling others to build on their work. Carbon reporting has no such direct reciprocal benefit for individual researchers. What evidence supports the claim that venue-level reporting requirements will translate into actual reductions in emissions, rather than simply generating data without changing research behavior?
- The paper acknowledges a tension between recommending against carbon as a review criterion and advocating for carbon-aware leaderboards, and proposes dual reporting and reference-grid normalization as mitigations. However, this normalization can introduce other distortions: it would make a highly energy-inefficient experiment in, say, Quebec appear comparable to an efficient one in a carbon intensive region.
- The paper focuses primarily on training carbon and argues that inference reporting is more domain-specific, but the argument that inference costs can exceed training costs within weeks of deployment is one of the paper's strongest justifications for urgency. Why are inference reporting standards not developed with the same level of detail as training standards in this proposal, and do the authors plan a separate follow-up proposal addressing inference, or should this paper be extended to cover it?

**Alternative Views Section:**

Yes

**Compliance With Llm Reviewing Policy A Conservative:**

Affirmed.

**Discussion Potential:**

3

**Final Justification:**

The rebuttal has addressed my concerns, I keep my original accept rating. I hope the promised discussions are included in the final paper. I think this can be a valuable and influential paper for the community.

**Paper Summary:**

In this position paper, the authors argue that standardized carbon footprint reporting should become a standardised routine element of ML research publications, in the same way we reoprt hardware specifications or hyperparameters. The authors argue that the current widespread omission of energy and GhG reporting is a scientific gap rather than only an environmental concern. Efficiency gains without energy metrics are argued to be incomplete, and the same experiments in different locations can yield carbon footprint differing by over one OOM, hindering reproducibility and cross-paper comparisons.

To solve this problem, this paper introduces 5 distinct reporting metrics: Total energy consumption, grid carbon intensity, power use effectiveness, compute region, and carbon emissions, which are aligned with the GhG protocol and other specifications. The paper surveys existing tools for monitoring energy consumption, and proposes a minimal yet comprehensive protocol that specifies what to measure, what to report, and how to report it. The paper outlines a 3-stage adoption timeline, following recent precedents such as mandatory code releases and the NeurIPS reproduciblity checklists, with concrete targets at eatch stage. The authors also tackle equity concerns, and close the paper with a call to venues, reviewers, authors, and institutions to coordinate, so that carbon awareness becomes a foundational element of responsible ML.

**Position:**

Yes

**Position In Title:**

Yes

**Related Work:**

4

**Strengths And Weaknesses:**

Strenghts
- The framing (carbon reporting is scientific good practice rather than environmental activism) is a strong rethorical move. The analogies drawn to physics are compelling, precise, and effective.
- The paper is very concrete, delivering precise actionable artifacts, from the standardised metrics, and ways to use them adequately. A conference or a researcher could use this paper directly.
- The adoption timeline in the paper is well designed, modelled expliclitly on previous success of ML community efforts towards transparency and reproducibility. The timeline is gradual to avoid backlash, and generally robust.
- The alternative views section is really solid and honest, with no strawmen arguments, and with real nuance.
- The distinction between training and inference in carbon emissions and energy consumption is really important, as many models only report training costs, which many times is a small fraction of all usage.
- Related work is really well covered and comprehensive.
- The paper is generally well written and easy to follow and was a genuine pleasure to read.

Weaknesses
- A central argument of this paper is that "carbon reporting is good" which is not really a controversial opinion that will merit a lot of discussion. The concrete actionable items are, however, much more likely to draw discussions.
- The proposed timeline is weak in terms of enforcement mechanisms. It simply expects that the ML community will adopt carbon footprint reporting as a community expectation. While this is possibly a strong-enough force, it may be insufficient if reviewers don't pressure for this explicitly.
- There is an argument to be made that carbon footprint monitoring is not that comparable to open source code release, as they tackle different problems and, in some cases, open source code has additional benefits beyond scientific good practices (Eg more visibility, citations, job opportunities), which may have created stronger forces towards it being standard practice than what carbon reporting may.
- The paper acknowledges the tension between equity and carbon footprint reporting but does not fully resolve it.
- The paper occasionally conflates scientific transparency with environmental impact reduction. These are related but distinct goals with  different implications for policy design. A reporting system optimized purely for transparency might look different from one optimized to reduce actual emissions.

**Support:**

4

---

> ### Author Rebuttal · Authors · 2026-03-26
>
> We thank Reviewer z2pR for the thoughtful and detailed review. On the distinction between transparency and impact reduction raised in the weaknesses: we agree these are related but distinct goals. Our proposal is primarily a transparency intervention; emission reductions are a hoped-for but not guaranteed downstream effect, and the paper's framing reflects this (Section 1, Section 7). The three questions below identify real tensions in our proposal, and we address each in turn.
>
> **Does reporting lead to reduction?** We acknowledge that direct causal evidence in ML is limited, and state this explicitly in Section 1. The strongest analogous precedent is reproducibility checklists: NeurIPS introduced them in 2019 and achieved widespread adoption within two years (Pineau et al., 2021), with measurable improvements in reporting quality. The mechanism is not that reporting directly reduces emissions, but that visibility changes what the community values. Once energy data is routinely available, three effects become possible: researchers can identify and adopt efficient methods, reviewers can contextualize massive compute for marginal gains (Section 5.1), and leaderboards can surface Pareto-optimal models (Section 4.2). We do not claim certainty that this chain will hold for carbon, and Section 1 states this epistemic limitation explicitly. We cannot rule out that reporting becomes a compliance exercise without behavioral change. However, even in that scenario, the accumulated data enables meta-analyses that are currently impossible, such as tracking whether the field is becoming more energy-efficient over time. Our survey of 1,000 ICML 2025 papers (see our response to Reviewer PEyd for methodology and full results) provides indirect but relevant evidence: compute metadata reporting grew from 17% (NeurIPS 2019) to 57.3% (ICML 2025) over six years, as community norms around reproducibility strengthened. This growth occurred without any formal mandate, driven primarily by evolving expectations and reproducibility checklists. Energy (9.0%) and carbon (5.9%) reporting today resemble compute reporting circa 2019, and our proposal aims to catalyze a similar trajectory through the same mechanism: clear expectations, templates, and gradual normalization.
>
> **Normalization distortion.** This is a genuine tradeoff we should have discussed more carefully. Reference-grid normalization answers a specific question: "how efficient is this method, controlling for location?" It is not meant to replace actual emissions reporting. A high-energy experiment in Quebec would show low carbon but high normalized carbon, making the inefficiency visible. The key is dual reporting (Section 6.2): actual emissions capture real-world impact, normalized values capture algorithmic efficiency, and readers need both to draw correct conclusions. We will add a concrete example in revision showing how dual reporting prevents the misinterpretation Reviewer z2pR describes.
>
> **Inference reporting depth.** The asymmetry is intentional. Training carbon is a bounded, one-time cost that the paper's author fully controls and can measure. Inference carbon accumulates across deployments that the author typically does not control, depends on serving infrastructure, batch sizes, and traffic patterns that vary across providers, and requires domain-specific benchmarks (energy per 1000 tokens vs. per 1000 images). Section 4.4 provides the principle that training and inference should be reported separately and outlines domain-specific benchmarks, but intentionally leaves inference standards less prescriptive because premature standardization risks locking in metrics that do not generalize across deployment contexts. A dedicated follow-up addressing inference-specific standards across deployment contexts is a natural next step, and we note this as future work in Section 8.
>
> We commit to the following revision: a concrete dual-reporting example in Section 6.2 addressing normalization distortion.

---

> > ### Author Rebuttal · Reviewer_z2pR · 2026-04-03
> >
> > I have read the rebuttal, that has fully addressed my concerns. I am keeping my Accept recommendation, I believe this paper can be valuable for the community.

---

### Official Review · Reviewer_PEyd · 2026-03-13

**Significance:** 2
**Argument Clarity:** 3
**Rating:** 4
**Confidence:** 3

**Questions:**

n/a

**Alternative Views Section:**

Yes

**Compliance With Llm Reviewing Policy A Conservative:**

Affirmed.

**Discussion Potential:**

2

**Final Justification:**

Concerns in the original review were well addressed, so I increased my score from 3 to 4.

**Paper Summary:**

The paper argues that ML papers should adopt standardized energy/carbon footprint reporting and that ML conference organizers play a key role in enforcing widespread adoption. The authors propose a specific standardization format that focuses on reporting five key metrics: total energy consumption, carbon emissions, grid carbon intensity, Power Usage Effectiveness (PUE), and the compute region. The paper provides actionable suggestions for ML conference organizers and ML researchers to reduce the awareness-action gap that leads to these carbon footprint-related metrics rarely being reported in ML papers. The authors emphasize that reporting of large carbon footprints should not be penalized in reviewing (except if it leads to marginal improvements), but rather that the non-disclosure of this information is the key issue to solve.

**Position:**

Yes

**Position In Title:**

Yes

**Related Work:**

2

**Strengths And Weaknesses:**

Strengths:
- The issue of whether/how to standardize carbon footprint reporting in ML papers is a very relevant topic. And while it's already been the topic of discussion, the paper has a point that there's a gap between awareness and action in this domain.
- The provided standardization on how to report carbon footprints is useful and could serve as a base for adoption in ML conferences. The suggested reporting/protocols are quite actionable.

Weaknesses:
- It's not a thought-provoking position, of only limited novelty. The core argument that ML papers should report carbon footprints has already been repeatedly made. As a result, it's unclear whether the paper would inspire follow-up discussion and how the awareness-action gap will be fixed through this paper.
- The paper states that *"Without energy and emissions metrics, efficiency claims lack scientific rigor"*. It would be useful to point to a few such examples where carbon footprint-related efficiencies are claimed but not substantiated with actual emissions metrics. This would strengthen the paper's reasoning.
- Water usage footprint is not discussed.
- The authors cite Henderson et al. 2020 as evidence for the lack of environmental footprint disclosures in ML papers and state that *"While awareness has grown since 2019, reporting remains the exception rather than the rule"*. This seems likely, but e.g., reporting of GPU-hours used has become standardized at venues like NeurIPS. Quantifying how much this awareness has changed and how large the remaining gap is would strengthen the paper's position.

**Support:**

3

---

> ### Author Rebuttal · Authors · 2026-03-30
>
> We thank Reviewer PEyd for the detailed assessment and for identifying concrete gaps that we can address substantively.
>
> **Novelty and discussion potential.** We agree the high-level claim ("ML should report carbon") is not new. Our contribution is not the observation but the operationalization: five standardized metrics aligned with the GHG Protocol and SCI specification, a measurement protocol specifying what to measure and how (Section 4.5), reporting templates at three tiers (Appendix B), and a phased timeline modeled on successful precedents. These artifacts are what distinguish a position paper that can be acted upon from one that restates a known concern. The discussion potential lies in the specific design choices, such as whether to normalize by reference grid (Section 6.2), how to handle privacy in double-blind review (raised by Reviewer zSFQ, addressed in our rebuttal), and whether inference standards should be prescriptive or domain-specific (raised by Reviewer z2pR). These are open questions where reasonable researchers disagree, and community discussion would benefit from a concrete proposal to react to.
>
> **Examples of unsubstantiated efficiency claims.** To quantify this gap, we conducted a large-scale survey of 1,000 randomly sampled ICML 2025 accepted papers (from 3,260 total). We performed automated keyword matching on full-text PDFs across three categories: compute metadata (e.g., GPU model names, FLOPs, training time), energy data (e.g., kWh, power consumption, CodeCarbon), and carbon data (e.g., CO2, carbon footprint, carbon emissions), with manual verification to remove false positives. Analysis code and per-paper results will be open-sourced upon acceptance. Among all 1,000 papers, 573 (57.3%) report compute metadata such as GPU types or FLOPs, often in the context of claiming efficiency gains over baselines. Yet only 90 (9.0%) report any energy data and only 59 (5.9%) mention carbon. This means that the vast majority of papers claiming "efficient" methods do so using only FLOP counts or wall-clock time, without energy validation. As noted in Section 3.2, FLOPs do not map linearly to energy: a model with 20% fewer FLOPs may consume the same energy due to memory access patterns and hardware utilization. We will add concrete examples from the survey illustrating papers that claim efficiency improvements based solely on FLOPs while omitting energy measurement.
>
> **Quantifying the current gap.** Reviewer PEyd correctly notes that our evidence relies on Henderson et al. (2020), which surveyed NeurIPS 2019. Our survey directly addresses this:
>
> |  | NeurIPS 2019 (n=100) | ICML 2025 (n=1,000) |
> |--|----------------------|-------------------|
> | Compute metadata | 17% | 57.3% |
> | Energy data | 1% | 9.0% |
> | Carbon data | 0% | 5.9% |
> | All three | N/A | 1.4% |
>
> Compute reporting has improved substantially (17% to 57.3%), but energy and carbon reporting remain in single digits despite six years of growing awareness, available tools, and high-profile disclosures from BLOOM, OPT, and Llama. This quantifies the awareness-action gap our paper identifies: awareness has increased, but standardized action has not followed.
>
> **Water usage.** This is a valid omission. Data center water consumption for cooling is a growing concern, with estimates that training GPT-3 consumed approximately 700,000 liters of freshwater (Li et al., "Making AI Less Thirsty," 2023). We will add water footprint as a recommended supplementary metric in the reporting guidelines, acknowledging that measurement tools are less mature than for energy and carbon.
>
> We commit to the following revisions: (1) ICML 2025 survey (n=1,000) with NeurIPS 2019 comparison table in Section 3, with analysis code and data open-sourced; (2) concrete examples of efficiency claims without energy data; (3) water usage as a supplementary reporting metric in Section 4.

---

> > ### Author Rebuttal · Reviewer_PEyd · 2026-04-03
> >
> > The rebuttal has addressed my concerns quite well.
> >
> > One follow-up: I had under-appreciated the notion that FLOPs do not necessarily correlate fully with energy usage (outside of trivial reasons like using different data centers/servers/locations), as described in Section 3.2. That section presents this as a fact, but no reference or example is provided (e.g., it's not clear where the 20% fewer flops but same energy usage example is coming from... is that magnitude a guess?). It would be more convincing to include at least one reference/example that illustrates this.

---

### Official Review · Reviewer_zSFQ · 2026-03-13

**Significance:** 3
**Argument Clarity:** 3
**Rating:** 5
**Confidence:** 4

**Questions:**

- Have the authors considered more incremental measures as a means to achieve proper climate reporting in the long term?  What would this look like between the stages of reporting simple metrics, such as GPU hours + hardware, and more detailed carbon metrics?
- Beyond identifying the issue and encouraging researchers to engage in this reporting, have the authors identified alternative incentives that may strengthen adoption?

**Alternative Views Section:**

No

**Compliance With Llm Reviewing Policy A Conservative:**

Affirmed.

**Discussion Potential:**

2

**Final Justification:**

The rebutall has addressed most of my concerns.

**Paper Summary:**

This paper argues for the need for carbon footprint-based reporting in machine learning (ML) research.  The authors discuss current gaps, i.e.,  the limited metrics reported (and how infrequently they are even reported), why reporting carbon-based metrics is important, existing tools that might be useful in reporting these, and urge the importance of this consideration in ML venues/reviewers/researchers to prioritize this.

**Position:**

Yes

**Position In Title:**

Yes

**Related Work:**

3

**Strengths And Weaknesses:**

### Strengths
- **Motivation**: Having better reporting to understand the carbon impact of ML research is an important consideration, especially with the significant impact modern large-scale models have on energy consumption.

### Weaknesses
- **Practical Implementation**: The authors discuss, firstly, how a minority of papers even report metrics such as computing time/GPU hours.  In my view, making this initial step a priority and urging venues/researchers to at least report it would be a much more practical suggestion.  While I agree with the authors that more reporting would be valuable, I believe their arguments for going beyond simple metrics like GPU-hours and hardware as a standard would not be practical, and that they haven't fully considered the major challenges that would be faced with metrics beyond this.
  - **Large-Scale Usage**: Tracking actual emissions with software- or hardware-based solutions is likely to be much more challenging.  For example, for users of compute clusters, it is unclear whether and how quickly system admins would prioritize this, and given the view of pushing venues/reviewers to consider these metrics, I see significant limitations in pushing for such measures.
  - **Reviewer Burden**: Compared to simple metrics, e.g., GPU-hours, understanding carbon costs is much less interpretable, so I believe the barrier for research to effectively understand these costs and contextualize them is non-trivial.  Perhaps this could be done at the venue level, but asking reviewers (who often have limited engagement in the review process for the suggested venues) is not practical.
  - **Privacy**: One aspect not touched on is the potential of privacy concerns related to reporting these metrics.  For example, reporting actual CO2 usage and grid-normalized CO2 usage may be enough to identify the authors' geographical locations.  If reporting metrics accessible to reviewers can in any way expose geographical location, these metrics would be a non-starter, as they'd pose issues with double-blind submissions.

Overall, this is an important issue that the ML community needs to consider more seriously, and the authors do a good job of outlining the reasons and offering insights into how reporting can be improved.  For example, the outline of existing software solutions to estimate carbon usage is great.  That said, some of the major adoption challenges are somewhat under-represented in my view, and a better focus on targeted, incremental steps that are more realistic in today's research environment, and on how we could progress from minor changes to more structural, long-term reporting, would be great.

**Support:**

2

---

> ### Author Rebuttal · Authors · 2026-03-26
>
> We thank Reviewer zSFQ for the constructive feedback, particularly for highlighting the practical adoption challenges and the privacy concern that strengthen the paper.
>
> **Incremental adoption path.** Reviewer zSFQ's intuition aligns with our design. Stage 1 of the timeline (Section 5.4) asks only for GPU-hours, hardware, and location, with no energy measurement at all. Measured kWh enters at Stage 2 after venues establish templates and community familiarity grows; full carbon accounting (Stage 3) applies only to large-scale experiments. To address the question of what the transition between stages looks like concretely: Stage 1 submissions that report hardware and location already contain enough information for venues to compute approximate carbon estimates at the aggregate level, which builds institutional familiarity before asking authors to measure energy themselves. This scaffolding effect is what makes Stage 2 realistic rather than a sudden jump. We recognize the current presentation may bury this point, and will restructure Section 5.4 to foreground how minimal the initial ask is and how each stage scaffolds the next. On cluster-level tracking specifically: when direct measurement is unavailable, our protocol accepts TDP-based approximations with acknowledged uncertainty (Section 6.1). SLURM clusters increasingly support job-level energy accounting, and cloud providers already offer carbon dashboards, but Stage 1 deliberately avoids depending on any of this.
>
> **Readiness for the next step.** Our survey of 1,000 ICML 2025 papers (methodology and full results in our response to Reviewer PEyd) supports this directly: compute metadata reporting has reached 57.3%, up from 17% at NeurIPS 2019. This suggests the initial step Reviewer zSFQ describes is largely accomplished, and the community is ready to move toward energy and carbon reporting. Meanwhile, energy reporting remains at 9.0% and carbon at 5.9%, confirming that the gap lies specifically between compute metadata (now common) and energy/carbon data (still rare). Stage 1 of our timeline builds directly on this existing norm, asking only for hardware and location, which 57.3% of papers already partially provide.
>
> **Reviewer burden.** Carbon data in our proposal is not a review criterion (Section 5.1). Reviewers check presence and basic plausibility, not whether a carbon cost is justified. This is the same role they play for reproducibility checklists today. We recognize, however, that carbon numbers are less intuitive than GPU-hours. To address this, Stage 2 (Section 5.4) envisions venues publishing empirical ranges for common experiment types (e.g., fine-tuning a 7B model typically consumes 50-200 kWh), giving reviewers a quick reference for plausibility without requiring expertise in carbon accounting. Deeper interpretation and cross-paper trend analysis belong at the venue level, not in individual reviews.
>
> **Privacy and double-blind review.** This is a genuine gap in our paper. Reporting grid intensity or compute region can narrow author identity, which conflicts with double-blind review. Two mitigations: carbon metadata could be submitted as supplementary information visible only to area chairs, not reviewers; and authors could report reference-grid normalized values (Section 6.2) during review, disclosing actual location only in the camera-ready. We will add a dedicated discussion in revision.
>
> **Incentives.** Reviewer zSFQ rightly asks what drives adoption beyond encouragement. Reproducibility checklists offer the closest precedent, achieving widespread adoption within two years through venue-level expectations alone (Pineau et al., 2021). Carbon-aware leaderboards (Section 4.2) add competitive incentive. A further driver we should have emphasized: as universities and funding agencies increasingly require sustainability reporting for grants and institutional audits, carbon data from publications becomes directly reusable, giving authors a practical reason to collect it. That said, carbon reporting lacks the direct reciprocal benefit of code release such as citations and visibility, and we should be more honest about this asymmetry in the paper.
>
> We commit to the following revisions: (1) restructured Section 5.4 foregrounding the minimal Stage 1 requirements; (2) discussion of privacy and double-blind considerations; (3) more candid treatment of the incentive gap compared to code release.

---

> > ### Author Rebuttal · Reviewer_zSFQ · 2026-04-04
> >
> > The authors have addressed most of my concerns.  I still believe that there will be major limitations and burdens in extending beyond GPU hours.  However, the authors offer some useful insights, and hopefully, as a community, we can work toward better reporting.  I've raised my score to accept since I believe it to be a paper that has strong potential of discussion among the community.

---

### Official Review · Reviewer_WcUw · 2026-03-14

**Significance:** 2
**Argument Clarity:** 3
**Rating:** 5
**Confidence:** 3

**Questions:**

See Weaknesses

**Alternative Views Section:**

Yes

**Compliance With Llm Reviewing Policy A Conservative:**

Affirmed.

**Discussion Potential:**

2

**Final Justification:**

I think the paper does a reasonable job of defending its position and as long as the final version is able to incorporate the changes committed in the rebuttal, I will maintain my positive score

**Paper Summary:**

The paper proposes mandatory carbon emission reporting for ML papers. The paper argues that without reporting of carbon emissions and energy consumption, a model's efficiency cannot be truly gauged. The paper proposes that every paper should report not only carbon emissions and energy consumption but also grid carbon intensity, power usage efficiency and the geographic location of compute. The paper also discusses the challenges in practical implementation of this idea and the alternative opinions.

**Position:**

Yes

**Position In Title:**

Yes

**Related Work:**

2

**Strengths And Weaknesses:**

Strengths:
1. The paper puts forward a very important idea to improve the trust in reliability of ML models. This will be of interest to the broader ML community.
2. The paper does a good job of discussing the alternative opinions giving a good overall perspective to the reader.

Weaknesses:
1. It would have been better if paper had put a little more details and analysis of the current situation of carbon reporting.
2. The paper should try to quantify the costs of energy consumption and carbon emissions explicitly to better support its case.

**Support:**

3

---

> ### Author Rebuttal · Authors · 2026-03-30
>
> We thank Reviewer WcUw for recognizing the importance of carbon reporting for ML reliability and for the constructive suggestions.
>
> **Current situation analysis.** To address this, we conducted a large-scale survey of 1,000 ICML 2025 papers (methodology and full results in our response to Reviewer PEyd). The results are striking: 57.3% report compute metadata, but only 9.0% mention energy and 5.9% mention carbon, with just 1.4% reporting all three. This updates Henderson et al.'s 2020 NeurIPS 2019 findings and confirms that while compute metadata has become commonplace, energy and carbon reporting remain rare. We will incorporate this survey with full methodology in revision and will open-source the analysis code and per-paper results upon acceptance.
>
> **Quantifying costs.** To make the economic and environmental costs explicit: training GPT-3 consumed approximately 1,287 MWh, emitting 552 tCO2eq (Patterson et al., "Carbon Emissions and Large Neural Network Training," 2021), equivalent to roughly $154K in electricity at US average rates. Training Llama-3.1 405B emitted 11,390 tCO2eq (Dubey et al., "The Llama 3 Herd of Models," 2024), comparable to the annual emissions of approximately 2,500 US households. At inference scale, a model serving 10M queries per day at approximately 0.005 kWh per query consumes roughly 18,250 MWh per year, potentially exceeding training costs within months (Luccioni et al., "Power Hungry Processing," 2024). Our survey shows that 91.0% of papers omit energy data entirely, meaning these costs remain invisible across the vast majority of published research. We will add a dedicated cost quantification subsection in revision.
>
> We commit to the following revisions: (1) ICML 2025 survey results (n=1,000) added to Section 3, with analysis code and data open-sourced; (2) explicit cost quantification with worked examples.

---

> > ### Author Rebuttal · Reviewer_WcUw · 2026-04-02
> >
> > I have read the rebuttal. I am maintaining my score

---

### Decision · Program_Chairs · 2026-04-30

**Decision:**

Accept (regular)

**Comment:**

The paper addresses an important and timely topic of reporting and standardizing carbon footprint metrics as routine practice in ML research. The paper provides compelling evidence to defend this position. This paper will help close the current gap in the field between awareness of the issue and adoption of practical solutions.